# MAKING NEURAL PROGRAMMING ARCHITECTURES GENERALIZE VIA RECURSION

**Jonathon Cai, Richard Shin, Dawn Song**
Department of Computer Science
University of California, Berkeley
Berkeley, CA 94720, USA
`{jonathon,ricshin,dawnsong}@cs.berkeley.edu`

## ABSTRACT

Empirically, neural networks that attempt to learn programs from data have exhibited poor generalizability. Moreover, it has traditionally been difficult to reason about the behavior of these models beyond a certain level of input complexity. In order to address these issues, we propose augmenting neural architectures with a key abstraction: recursion. As an application, we implement recursion in the Neural Programmer-Interpreter framework on four tasks: grade-school addition, bubble sort, topological sort, and quicksort. We demonstrate superior generalizability and interpretability with small amounts of training data. Recursion divides the problem into smaller pieces and drastically reduces the domain of each neural network component, making it tractable to prove guarantees about the overall system's behavior. Our experience suggests that in order for neural architectures to robustly learn program semantics, it is necessary to incorporate a concept like recursion.

## 1 INTRODUCTION

Training neural networks to synthesize robust programs from a small number of examples is a challenging task. The space of possible programs is extremely large, and composing a program that performs robustly on the infinite space of possible inputs is difficult—in part because it is impractical to obtain enough training examples to easily disambiguate amongst all possible programs. Nevertheless, we would like the model to quickly learn to represent the right semantics of the underlying program from a small number of training examples, not an exhaustive number of them.

Thus far, to evaluate the efficacy of neural models on programming tasks, the only metric that has been used is generalization of expected behavior to inputs of greater complexity (Vinyals et al. (2015), Kaiser & Sutskever (2015), Reed & de Freitas (2016), Graves et al. (2016), Zaremba et al. (2016)). For example, for the addition task, the model is trained on short inputs and then tested on its ability to sum inputs with much longer numbers of digits. Empirically, existing models suffer from a common limitation—generalization becomes poor beyond a threshold level of complexity. Errors arise due to undesirable and uninterpretable dependencies and associations the architecture learns to store in some high-dimensional hidden state. This makes it difficult to reason about what the model will do when given complex inputs.

One common strategy to improve generalization is to use curriculum learning, where the model is trained on inputs of gradually increasing complexity. However, models that make use of this strategy eventually fail after a certain level of complexity (e.g. the single-digit multiplication task in Zaremba et al. (2016), the bubble sort task in Reed & de Freitas (2016), and the graph tasks in Graves et al. (2016)). In this version of curriculum learning, even though the inputs are gradually becoming more complex, the semantics of the program is succinct and does not change. Although the model is exposed to more and more data, it might learn spurious and overly complex representations of the program, as suggested in Zaremba et al. (2016). That is to say, the network does not learn the true program semantics.

In this paper, we propose to resolve these issues by explicitly incorporating recursion into neural architectures. Recursion is an important concept in programming languages and a critical tool to

reduce the complexity of programs. We find that recursion makes it easier for the network to learn the right program and generalize to unknown situations. Recursion enables provable guarantees on neural programs' behavior without needing to exhaustively enumerate all possible inputs to the programs. This paper is the first (to our knowledge) to investigate the important problem of provable generalization properties of neural programs. As an application, we incorporate recursion into the Neural Programmer-Interpreter architecture and consider four sample tasks: grade-school addition, bubble sort, topological sort, and quicksort. Empirically, we observe that the learned recursive programs solve all valid inputs with 100% accuracy after training on a very small number of examples, out-performing previous generalization results. Given verification sets that cover all the base cases and reduction rules, we can provide proofs that these learned programs generalize perfectly. This is the first time one can provide provable guarantees of perfect generalization for neural programs.

## 2 THE PROBLEM AND OUR APPROACH

### 2.1 THE PROBLEM OF GENERALIZATION

When constructing a neural network for the purpose of learning a program, there are two orthogonal aspects to consider. The first is the actual model architecture. Numerous models have been proposed for learning programs; to name a few, this includes the Differentiable Neural Computer (Graves et al., 2016), Neural Turing Machine (Graves et al., 2014), Neural GPU (Kaiser & Sutskever, 2015), Neural Programmer (Neelakantan et al., 2015), Pointer Network (Vinyals et al., 2015), Hierarchical Attentive Memory (Andrychowicz & Kurach, 2016), and Neural Random Access Machine (Kurach et al., 2016). The architecture usually possesses some form of memory, which could be internal (such as the hidden state of a recurrent neural network) or external (such as a discrete "scratch pad" or a memory block with differentiable access). The second is the training procedure, which consists of the form of the training data and the optimization process. Almost all architectures train on program input/output pairs. The only model, to our knowledge, that does not train on input-output pairs is the Neural Programmer-Interpreter (Reed & de Freitas, 2016), which trains on synthetic execution traces.

To evaluate a neural network that learns a neural program to accomplish a certain task, one common evaluation metric is how well the learned model $M$ generalizes. More specifically, when $M$ is trained on simpler inputs, such as inputs of a small length, the *generalization* metric evaluates how well $M$ will do on more complex inputs, such as inputs of much longer length. $M$ is considered to have perfect generalization if $M$ can give the right answer for any input, such as inputs of arbitrary length.

As mentioned in Section 1, all approaches to neural programming today fare poorly on this generalization issue. We hypothesize that the reason for this is that the neural network learns to spuriously depend on specific characteristics of the training examples that are irrelevant to the true program semantics, such as length of the training inputs, and thus fails to generalize to more complex inputs.

In addition, none of the current approaches to neural programming provide a method or even aim to enable provable guarantees about generalization. The memory updates of these neural programs are so complex and interdependent that it is difficult to reason about the behaviors of the learned neural program under previously unseen situations (such as problems with longer inputs). This is highly undesirable, since being able to provide the correct answer in all possible settings is one of the most important aspects of any learned neural program.

### 2.2 OUR APPROACH USING RECURSION

In this paper, we propose that the key abstraction of *recursion* is necessary for neural programs to generalize. The general notion of recursion has been an important concept in many domains, including mathematics and computer science. In computer science, recursion (as opposed to iteration) involves solving a larger problem by combining solutions to smaller instances of the same problem. Formally, a function exhibits recursive behavior when it possesses two properties: (1) Base cases— terminating scenarios that do not use recursion to produce answers; (2) A set of rules that reduces all other problems toward the base cases. Some functional programming languages go so far as not to define any looping constructs but rely solely on recursion to enable repeated execution of the same code.

In this paper, we propose that recursion is an important concept for neural programs as well. In fact, we argue that recursion is an essential element for neural programs to generalize, and makes it tractable to prove the generalization of neural programs. Recursion can be implemented differently for different neural programming models. Here as a concrete and general example, we consider a general Neural Programming Architecture (NPA), similar to Neural Programmer-Interpreter (NPI) in Reed & de Freitas (2016). In this architecture, we consider a core controller, e.g., an LSTM in NPI's case, but possibly other networks in different cases. There is a (changing) list of neural programs used to accomplish a given task. The core controller acts as a dispatcher for the programs. At each time step, the core controller can decide to select one of the programs to call with certain arguments. When the program is called, the current context including the caller's memory state is stored on a stack; when the program returns, the stored context is popped off the stack to resume execution in the previous caller's context.

In this general Neural Programming Architecture, we show it is easy to support recursion. In particular, recursion can be implemented as a program calling itself. Because the context of the caller is stored on a stack when it calls another program and the callee starts in a fresh context, this enables recursion simply by allowing a program to call itself. In practice, we can additionally use tail recursion optimization to avoid problems with the call stack growing too deep. Thus, any general Neural Programming Architecture supporting such a call structure can be made to support recursion. In particular, this condition is satisfied by NPI, and thus the NPI model naturally supports recursion (even though the authors of NPI did not consider this aspect explicitly).

By nature, recursion reduces the complexity of a problem to simpler instances. Thus, recursion helps decompose a problem and makes it easier to reason about a program's behavior for previously unseen situations such as longer inputs. In particular, given that a recursion is defined by two properties as mentioned before, the base cases and the set of reduction rules, we can prove a recursive neural program generalizes perfectly if we can prove that (1) it performs correctly on the base cases; (2) it learns the reduction rules correctly. For many problems, the base cases and reduction rules usually consist of a finite (often small) number of cases. For problems where the base cases may be extremely large or infinite, such as certain forms of motor control, recursion can still help reduce the problem of generalization to these two aspects and make the generalization problem significantly simpler to handle and reason about.

As a concrete instantiation, we show in this paper that we can enable recursive neural programs in the NPI model, and thus enable perfectly generalizable neural programs for tasks such as sorting where the original, non-recursive NPI program fails. As aforementioned, the NPI model naturally supports recursion. However, the authors of NPI did not consider explicitly the notion of recursion and as a consequence, did not learn recursive programs. We show that by modifying the training procedure, we enable the NPI model to learn recursive neural programs. As a consequence, our learned neural programs empirically achieve perfect generalization from a very small number of training examples. Furthermore, given a verification input set that covers all base cases and reduction rules, we can formally prove that the learned neural programs achieve perfect generalization after verifying its behavior on the verification input set. This is the first time one can provide provable guarantees of perfect generalization for neural programs.

We would also like to point out that in this paper, we provide as an example one way to train a recursive neural program, by providing a certain training execution trace to the NPI model. However, our concept of recursion for neural programs is general. In fact, it is one of our future directions to explore new ways to train a recursive neural program without providing explicit training execution traces or with only partial or non-recursive traces.

## 3 APPLICATION TO LEARNING RECURSIVE NEURAL PROGRAMS WITH NPI

### 3.1 BACKGROUND: NPI ARCHITECTURE

As discussed in Section 2, the Neural Programmer-Interpreter (NPI) is an instance of a Neural Programmer Architecture and hence it naturally supports recursion. In this section, we give a brief review of the NPI architecture from Reed & de Freitas (2016) as background.

We describe the details of the NPI model relevant to our contributions. We adapt machinery from the original paper slightly to fit our needs. The NPI model has three learnable components: a task-agnostic core, a program-key embedding, and domain-specific encoders that allow the NPI to operate in diverse environments.

The NPI accesses an external environment, $Q$, which varies according to the task. The core module of the NPI is an LSTM controller that takes as input a slice of the current external environment, via a set of pointers, and a program and arguments to execute. NPI then outputs the return probability and next program and arguments to execute. Formally, the NPI is represented by the following set of equations:

$$s_t = f_{enc}(e_t, a_t)$$

$$h_t = f_{lstm}(s_t, p_t, h_{t-1})$$

$$r_t = f_{end}(h_t), p_{t+1} = f_{prog}(h_t), a_{t+1} = f_{arg}(h_t)$$

$t$ is a subscript denoting the time-step; $f_{enc}$ is a domain-specific encoder (to be described later) that takes in the environment slice $e_t$ and arguments $a_t$; $f_{lstm}$ represents the core module, which takes in the state $s_t$ generated by $f_{enc}$, a program embedding $p_t \in \mathbb{R}^P$, and hidden LSTM state $h_t$; $f_{end}$ decodes the return probability $r_t$; $f_{prog}$ decodes a program key embedding $p_{t+1}$;[1] and $f_{arg}$ decodes arguments $a_{t+1}$. The outputs $r_t, p_{t+1}, a_{t+1}$ are used to determine the next action, as described in Algorithm 1. If the program is primitive, the next environmental state $e_{t+1}$ will be affected by $p_t$ and $a_t$, i.e. $e_{t+1} \sim f_{env}(e_t, p_t, a_t)$. As with the original NPI architecture, the experiments for this paper always used a 3-tuple of integers $a_t = (a_t(1), a_t(2), a_t(3))$.

---

**Algorithm 1** Neural programming inference

1: **Inputs**: Environment observation $e$, program $p$, arguments $a$, stop threshold $\alpha$
2: **function** RUN($e, p, a$)
3:　　$h \leftarrow \mathbf{0}, r \leftarrow 0$
4:　　**while** $r < \alpha$ **do**
5:　　　　$s \leftarrow f_{enc}(e, a), h \leftarrow f_{lstm}(s, p, h)$
6:　　　　$r \leftarrow f_{end}(h), p_2 \leftarrow f_{prog}(h), a_2 \leftarrow f_{arg}(h)$
7:　　　　**if** $p$ is a primitive function **then**
8:　　　　　　$e \leftarrow f_{env}(e, p, a)$.
9:　　　　**else**
10:　　　　　　**function** RUN($e, p_2, a_2$)

---

A description of the inference procedure is given in Algorithm 1. Each step during an execution of the program does one of three things: (1) another subprogram along with associated arguments is called, as in Line 10, (2) the program writes to the environment if it is primitive, as in Line 8, or (3) the loop is terminated if the return probability exceeds a threshold $\alpha$, after which the stack frame is popped and control is returned to the caller. In all experiments, $\alpha$ is set to 0.5. Each time a subprogram is called, the stack depth increases.

The training data for the Neural Programmer-Interpreter consists of full execution traces for the program of interest. A single element of an execution trace consists of a step input-step output pair, which can be synthesized from Algorithm 1: this corresponds to, for a given time-step, the step input tuple $(e, p, a)$ and step output tuple $(r, p_2, a_2)$. An example of part of an addition task trace, written in shorthand, is given in Figure 1. For example, a step input-step output pair in Lines 2 and 3 of the left-hand side of Figure 1 is (ADD1, WRITE OUT 1). In this pair, the step input runs a subprogram ADD1 that has no arguments, and the step output contains a program WRITE that has arguments of OUT and 1. The environment and return probability are omitted for readability. Indentation indicates the stack is one level deeper than before.

It is important to emphasize that at inference time in the NPI, the hidden state of the LSTM controller is reset (to zero) at each subprogram call, as in Line 3 of Algorithm 1 ($h \leftarrow \mathbf{0}$). This functionality

---

[1]The original NPI paper decodes to a program key embedding $k_t \in \mathbb{R}^K$ and then computes a program embedding $p_{t+1}$, which we also did in our implementation, but we omit this for brevity.

|  | Non-Recursive |  | Recursive |
|---|---|---|---|
| 1 | ADD | 1 | ADD |
| 2 | ADD1 | 2 | ADD1 |
| 3 | WRITE OUT 1 | 3 | WRITE OUT 1 |
| 4 | CARRY | 4 | CARRY |
| 5 | PTR CARRY LEFT | 5 | PTR CARRY LEFT |
| 6 | WRITE CARRY 1 | 6 | WRITE CARRY 1 |
| 7 | PTR CARRY RIGHT | 7 | PTR CARRY RIGHT |
| 8 | LSHIFT | 8 | LSHIFT |
| 9 | PTR INP1 LEFT | 9 | PTR INP1 LEFT |
| 10 | PTR INP2 LEFT | 10 | PTR INP2 LEFT |
| 11 | PTR CARRY LEFT | 11 | PTR CARRY LEFT |
| 12 | PTR OUT LEFT | 12 | PTR OUT LEFT |
| 13 | ADD1 | 13 | **ADD** |
| 14 | ... | 14 | ... |

Figure 1: Addition Task. The non-recursive trace loops on cycles of ADD1 and LSHIFT, whereas in the recursive version, the ADD function calls itself (bolded).

is critical for implementing recursion, since it permits us to restrict our attention to the currently relevant recursive call, ignoring irrelevant details about other contexts.

## 3.2 RECURSIVE FORMULATIONS FOR NPI PROGRAMS

We emphasize the overall goal of this work is to enable the learning of a recursive program. The learned recursive program is different from neural programs learned in all previous work in an important aspect: previous approaches do not explicitly incorporate this abstraction, and hence generalize poorly, whereas our learned neural programs incorporate recursion and achieve perfect generalization.

Since NPI naturally supports the notion of recursion, a key question is how to enable NPI to learn recursive programs. We found that changing the NPI training traces is a simple way to enable this. In particular, we construct new training traces which explicitly contain recursive elements and show that with this type of trace, NPI easily learns recursive programs. In future work, we would like to decrease supervision and construct models that are capable of coming up with recursive abstractions themselves.

In what follows, we describe the way in which we constructed NPI training traces so as to make them contain recursive elements and thus enable NPI to learn recursive programs. We describe the recursive re-formulation of traces for two tasks from the original NPI paper—grade-school addition and bubble sort. For these programs, we re-use the appropriate program sets (the associated subprograms), and we refer the reader to the appendix of Reed & de Freitas (2016) for further details on the subprograms used in addition and bubble sort. Finally, we implement recursive traces for our own topological sort and quicksort tasks.

**Grade School Addition.** For grade-school addition, the domain-specific encoder is

$$f_{enc}(Q, i_1, i_2, i_3, i_4, a_t) = MLP([Q(1, i_1), Q(2, i_2), Q(3, i_3), Q(4, i_4), a_t(1), a_t(2), a_t(3)]),$$

where the environment $Q \in \mathbb{R}^{4 \times N \times K}$ is a scratch-pad that contains four rows (the first input number, the second input number, the carry bits, and the output) and $N$ columns. $K$ is set to 11, to represent the range of 10 possible digits, along with a token representing the end of input.[2] At any given time, the NPI has access to values pointed to by four pointers in each of the four rows, represented by $Q(1, i_1), Q(2, i_2), Q(3, i_3)$, and $Q(4, i_4)$.

The non-recursive trace loops on cycles of ADD1 and LSHIFT. ADD1 is a subprogram that adds the current column (writing the appropriate digit to the output row and carrying a bit to the next column if needed). LSHIFT moves the four pointers to the left, to move to the next column. The program terminates when seeing no numbers in the current column.

Figure 1 shows examples of non-recursive and recursive addition traces. We make the trace recursive by adding a tail recursive call into the trace for the ADD program after calling ADD1 and LSHIFT,

---

[2]The original paper uses $K = 10$, but we found it necessary to augment the range with an end token, in order to terminate properly.

Non-Recursive

```
1   BUBBLESORT
2     BUBBLE
3       PTR 2 RIGHT
4       BSTEP
5         COMPSWAP
6
7         RSHIFT
8           PTR 1 RIGHT
9           PTR 2 RIGHT
10        BSTEP
11          COMPSWAP
12            SWAP 1 2
13          RSHIFT
14            PTR 1 RIGHT
15            PTR 2 RIGHT
16    RESET
17      LSHIFT
18        PTR 1 LEFT
19        PTR 2 LEFT
20      LSHIFT
21        PTR 1 LEFT
22        PTR 2 LEFT
23      PTR 3 RIGHT
24    BUBBLE
25      ...
```

Partial Recursive

```
1   BUBBLESORT
2     BUBBLE
3       PTR 2 RIGHT
4       BSTEP
5         COMPSWAP
6
7         RSHIFT
8           PTR 1 RIGHT
9           PTR 2 RIGHT
10        BSTEP
11          COMPSWAP
12            SWAP 1 2
13          RSHIFT
14            PTR 1 RIGHT
15            PTR 2 RIGHT
16    RESET
17      LSHIFT
18        PTR 1 LEFT
19        PTR 2 LEFT
20      LSHIFT
21        PTR 1 LEFT
22        PTR 2 LEFT
23      PTR 3 RIGHT
24    BUBBLESORT
25      BUBBLE
26        ...
```

Full Recursive

```
1   BUBBLESORT
2     BUBBLE
3       PTR 2 RIGHT
4       BSTEP
5         COMPSWAP
6
7         RSHIFT
8           PTR 1 RIGHT
9           PTR 2 RIGHT
10        BSTEP
11          COMPSWAP
12            SWAP 1 2
13          RSHIFT
14            PTR 1 RIGHT
15            PTR 2 RIGHT
16          BSTEP
17    RESET
18      LSHIFT
19        PTR 1 LEFT
20        PTR 2 LEFT
21        LSHIFT
22          PTR 1 LEFT
23          PTR 2 LEFT
24          LSHIFT
25      PTR 3 RIGHT
26    BUBBLESORT
27      BUBBLE
28        ...
```

Figure 2: Bubble Sort Task. The non-recursive trace loops on cycles of BUBBLE and RESET. The difference between the partial recursive and full recursive versions is in the indentation of Lines 10-15 and 20-22 (bolded), since in the full recursive version, BSTEP and LSHIFT are made tail recursive; the final calls to BSTEP and LSHIFT return immediately as they occur after the pointer reaches the end of the array. Also note that COMPSWAP conditionally swaps numbers under the bubble pointers.

as in Line 13 of the right-hand side of Figure 1. Via the recursive call, we effectively forget that the column just added exists, since the recursive call to ADD starts with a new hidden state for the LSTM controller. Consequently, there is no concept of length relevant to the problem, which has traditionally been an important focus of length-based curriculum learning.

**Bubble Sort.** For bubble sort, the domain-specific encoder is

$$f_{enc}(Q, i_1, i_2, i_3, a_t) = MLP([Q(1, i_1), Q(1, i_2), i_3 == length, a_t(1), a_t(2), a_t(3)]),$$

where the environment $Q \in \mathbb{R}^{1 \times N \times K}$ is a scratch-pad that contains 1 row, to represent the state of the array as sorting proceeds in-place, and $N$ columns. $K$ is set to 11, to denote the range of possible numbers (0 through 9), along with the start/end token (represented with the same encoding) which is observed when a pointer reaches beyond the bounds of the input. At any given time, the NPI has access to the values referred to by two pointers, represented by $Q(1, i_1)$ and $Q(1, i_2)$,. The pointers at index $i_1$ and $i_2$ are used to compare the pair of numbers considered during the bubble sweep, swapping them if the number at $i_1$ is greater than that in $i_2$. These pointers are referred to as bubble pointers. The pointer at index $i_3$ represents a counter internal to the environment that is incremented once after each pass of the algorithm (one cycle of BUBBLE and RESET); when incremented a number of times equal to the length of the array, the flag $i_3 == length$ becomes true and terminates the entire algorithm .

The non-recursive trace loops on cycles of BUBBLE and RESET, which logically represents one bubble sweep through the array and reset of the two bubble pointers to the very beginning of the array, respectively. In this version, there is a dependence on length: BSTEP and LSHIFT are called a number of times equivalent to one less than the length of the input array, in BUBBLE and RESET respectively.

Inside BUBBLE and RESET, there are two operations that can be made recursive. BSTEP, used in BUBBLE, compares pairs of numbers, continuously moving the bubble pointers once to the right each time until reaching the end of the array. LSHIFT, used in RESET, shifts the pointers left until reaching the start token.

We experiment with two levels of recursion—partial and full. Partial recursion only adds a tail recursive call to BUBBLESORT after BUBBLE and RESET, similar to the tail recursive call described previously for addition. The partial recursion is not enough for perfect generalization, as will be presented later in Section 4. Full recursion, in addition to making the aforementioned tail recursive call, adds two additional recursive calls; BSTEP and LSHIFT are made tail recursive. Figure 2 shows examples of traces for the different versions of bubble sort. Training on the full recursive trace leads to perfect generalization, as shown in Section 4. We performed experiments on the partially recursive version in order to examine what happens when only one recursive call is implemented, when in reality three are required for perfect generalization.

---

**Algorithm 2** Depth First Search Topological Sort

---

1: Color all vertices white.
2: Initialize an empty stack $S$ and a directed acyclic graph $DAG$ to traverse.
3: Begin traversing from Vertex 1 in the DAG.
4: **function** TOPOSORT($DAG$)
5: **while** there is still a white vertex $u$: **do**
6: color[$u$] = grey
7: $v_{active} = u$
8: **do**
9: **if** $v_{active}$ has a white child $v$ **then**
10: color[$v$] = grey
11: push $v_{active}$ onto $S$
12: $v_{active} = v$
13: **else**
14: color[$v_{active}$] = black
15: Write $v_{active}$ to result
16: **if** $S$ is empty **then** pass
17: **else** pop the top vertex off $S$ and set it to $v_{active}$
18: **while** $S$ is not empty

---

**Topological Sort.** We choose to implement a topological sort task for graphs. A *topological sort* is a linear ordering of vertices such that for every directed edge $(u, v)$ from $u$ to $v$, $u$ comes before $v$ in the ordering. This is possible if and only if the graph has no directed cycles; that is to say, it must be a directed acyclic graph (DAG). In our experiments, we only present DAG's as inputs and represent the vertices as values ranging from $1, \ldots, n$, where the DAG contains $n$ vertices.

Directed acyclic graphs are structurally more diverse than inputs in the two tasks of grade-school addition and bubble sort. The degree for any vertex in the DAG is variable. Also the DAG can have potentially more than one connected component, meaning it is necessary to transition between these components appropriately.

Algorithm 2 shows the topological sort task of interest. This algorithm is a variant of depth first search. We created a program set that reflects the semantics of Algorithm 2. For brevity, we refer the reader to the appendix for further details on the program set and non-recursive and recursive trace-generating functions used for topological sort.

For topological sort, the domain-specific encoder is

$$f_{enc}(DAG, Q_{color}, p_{stack}, p_{start}, v_{active}, childList, a_t)$$
$$= MLP([Q_{color}(p_{start}), Q_{color}(DAG[v_{active}][childList[v_{active}]]), p_{stack} == 1, a_t(1), a_t(2), a_t(3)]),$$

where $Q_{color} \in \mathbb{R}^{U \times 4}$ is a scratch-pad that contains $U$ rows, each containing one of four colors (white, gray, black, invalid) with one-hot encoding. $U$ varies with the number of vertices in the graph. We further have $Q_{result} \in \mathbb{N}^U$, a scratch-pad which contains the sorted list of vertices at the end of execution, and $Q_{stack} \in \mathbb{N}^U$, which serves the role of the stack $S$ in Algorithm 2. The contents of $Q_{result}$ and $Q_{stack}$ are not exposed directly through the domain-specific encoder; rather, we define primitive functions which manipulate these scratch-pads.

The DAG is represented as an adjacency list where $DAG[i][j]$ refers to the $j$-th child of vertex $i$. There are 3 pointers ($p_{result}, p_{stack}, p_{start}$), $p_{result}$ points to the next empty location in $Q_{result}$,

$p_{stack}$ points to the top of the stack in $Q_{stack}$, and $p_{start}$ points to the candidate starting node for a connected component. There are 2 variables ($v_{active}$ and $v_{save}$); $v_{active}$ holds the active vertex (as in Algorithm 2) and $v_{save}$ holds the value of $v_{active}$ before executing Line 12 of Algorithm 2. $childList \in \mathbb{N}^U$ is a vector of pointers, where $childList[i]$ points to the next child under consideration for vertex $i$.

The three environment observations aid with control flow in Algorithm 2. $Q_{color}(p_{start})$ contains the color of the current start vertex, used in the evaluation of the condition in the WHILE loop in Line 5 of Algorithm 2. $Q_{color}(DAG[v_{active}][childList[v_{active}]])$ refers to the color of the next child of $v_{active}$, used in the evaluation of the condition in the IF branch in Line 9 of Algorithm 2. Finally, the boolean $p_{stack} == 1$ is used to check whether the stack is empty in Line 18 of Algorithm 2.

An alternative way of representing the environment slice is to expose the values of the absolute vertices to the model; however, this makes it difficult to scale the model to larger graphs, since large vertex values are not seen during training time.

We refer the reader to the appendix for the non-recursive trace generating functions. In the non-recursive trace, there are four functions that can be made recursive—TOPOSORT, CHECK_CHILD, EXPLORE, and NEXT_START, and we add a tail recursive call to each of these functions in order to make the recursive trace. In particular, in the EXPLORE function, adding a tail recursive call resets and stores the hidden states associated with vertices in a stack-like fashion. This makes it so that we only need to consider the vertices in the subgraph that are currently relevant for computing the sort, allowing simpler reasoning about behavior for large graphs. The sequence of primitive operations (MOVE and WRITE operations) for the non-recursive and recursive versions are exactly the same.

**Quicksort.** We implement a quicksort task, in order to demonstrate that recursion helps with learning divide-and-conquer algorithms. We use the Lomuto partition scheme; the logic for the recursive trace is shown in Algorithm 3. For brevity, we refer the reader to the appendix for information about the program set and non-recursive and recursive trace-generating functions for quicksort. The logic for the non-recursive trace is shown in Algorithm 4 in the appendix.

---

**Algorithm 3** Recursive Quicksort

---

 1: Initialize an array $A$ to sort.
 2: Initialize $lo$ and $hi$ to be 1 and $n$, where $n$ is the length of $A$.
 3:
 4: **function** QUICKSORT($A, lo, hi$)
 5: **if** $lo < hi$: **then**
 6: p = PARTITION($A, lo, hi$)
 7: QUICKSORT($A, lo, p - 1$)
 8: QUICKSORT($A, p + 1, hi$)
 9:
10: **function** PARTITION($A, lo, hi$)
11: $pivot = lo$
12: **for** $j \in [lo, hi - 1]$ : **do**
13: **if** $A[j] \leq A[hi]$ **then**
14: swap $A[pivot]$ with $A[j]$
15: $pivot = pivot + 1$
16: swap $A[pivot]$ with $A[hi]$
17: return $pivot$

---

For quicksort, the domain-specific encoder is

$$f_{enc}(Q_{array}, Q_{stackLo}, Q_{stackHi}, p_{lo}, p_{hi}, p_{stackLo}, p_{stackHi}, p_{pivot}, p_j, a_t) =$$
$$MLP([Q_{array}(p_j) \leq Q_{array}(p_{hi}), p_j == p_{hi},$$
$$Q_{stackLo}(p_{stackLo} - 1) < Q_{stackHi}(p_{stackHi} - 1), p_{stackLo} == 1, a_t(1), a_t(2), a_t(3)]),$$

where $Q_{array} \in \mathbb{R}^{U \times 11}$ is a scratch-pad that contains $U$ rows, each containing one of 11 values (one of the numbers 0 through 9 or an invalid state). Our implementation uses two stacks $Q_{stackLo}$ and

$Q_{stackHi}$, each in $\mathbb{R}^U$, that store the arguments to the recursive QUICKSORT calls in Algorithm 3; before each recursive call, the appropriate arguments are popped off the stack and written to $p_{lo}$ and $p_{hi}$.

There are 6 pointers ($p_{lo}, p_{hi}, p_{stackLo}, p_{stackHi}, p_{pivot}, p_j$). $p_{lo}$ and $p_{hi}$ point to the $lo$ and $hi$ indices of the array, as in Algorithm 3. $p_{stackLo}$ and $p_{stackHi}$ point to the top (empty) positions in $Q_{stackLo}$ and $Q_{stackHi}$. $p_{pivot}$ and $p_j$ point to the $pivot$ and $j$ indices of the array, used in the PARTITION function in Algorithm 3. The 4 environment observations aid with control flow; $Q_{stackLo}(p_{stackLo} - 1) < Q_{stackHi}(p_{stackHi} - 1)$ implements the $lo < hi$ comparison in Line 5 of Algorithm 3, $p_{stackLo} == 1$ checks if the stacks are empty in Line 18 of Algorithm 4, and the other observations (all involving $p_{pivot}$ or $p_j$) deal with logic in the PARTITION function.

Note that the recursion for quicksort is not purely tail recursive and therefore represents a more complex kind of recursion that is harder to learn than in the previous tasks. Also, compared to the bubble pointers in bubble sort, the pointers that perform the comparison for quicksort (the COMP-SWAP function) are usually not adjacent to each other, making quicksort less local than bubble sort. In order to compensate for this, $p_{pivot}$ and $p_j$ require special functions (MOVE_PIVOT_LO and MOVE_J_LO) to properly set them to $lo$ in Lines 11 and 12 of the PARTITION function in Algorithm 3.

### 3.3 PROVABLY PERFECT GENERALIZATION

We show that if we incorporate recursion, the learned NPI programs can achieve provably perfect generalization for different tasks. Provably perfect generalization implies the model will behave correctly, given any valid input. In order to claim a proof, we must verify the model produces correct behavior over all base cases and reductions, as described in Section 2.

We propose and describe our verification procedure. This procedure verifies that all base cases and reductions are handled properly by the model via explicit tests. Note that recursion helps make this process tractable, because we only need to test a finite number of inputs to show that the model will work correctly on inputs of unbounded complexity. This verification phase only needs to be performed once after training.

Formally, verification consists of proving the following theorem:

$$\forall i \in V, M(i) \Downarrow P(i)$$

where $i$ denotes a sequence of step inputs (within one function call), $V$ denotes the set of valid sequences of step inputs, $M$ denotes the neural network model, $P$ denotes the correct program, and $P(i)$ denotes the next step output from the correct program. The arrow in the theorem refers to evaluation, as in big-step semantics. The theorem states that for the same sequence of step inputs, the model produces the exact same step output as the target program it aims to learn. $M$, as described in Algorithm 1, processes the sequence of step inputs by using an LSTM.

Recursion drastically reduces the number of configurations we need to consider during the verification phase and makes the proof tractable, because it introduces structure that eliminates infinitely long sequences of step inputs that would otherwise need to be considered. For instance, for recursive addition, consider the family $F$ of addition problems $a_n a_{n-1} \ldots a_1 a_0 + b_n b_{n-1} \ldots b_1 b_0$ where no CARRY operations occur. We prove every member of $F$ is added properly, given that subproblems $S = \{a_n a_{n-1} + b_n b_{n-1}, a_{n-1} a_{n-2} + b_{n-1} b_{n-2}, \ldots, a_1 a_0 + b_1 b_0\}$ are added properly.

Without using a recursive program, such a proof is not possible, because the non-recursive program runs on an arbitrarily long addition problem that creates correspondingly long sequences of step inputs; in the non-recursive formulation of addition, ADD calls ADD1 a number of times that is dependent on the length of the input. The core LSTM module's hidden state is preserved over all these ADD1 calls, and it is difficult to interpret with certainty what happens over longer timesteps without concretely evaluating the LSTM with an input of that length. In contrast, each call to the recursive ADD always runs for a fixed number of steps, even on arbitrarily long problems in $F$, so we can test that it performs correctly on a small, fixed number of step input sequences. This guarantees that the step input sequences considered during verification contain all step input sequences which arise during execution of an unseen problem in $F$, leading to generalization to any problem in $F$. Hence, if all subproblems in $S$ are added correctly, we have proven that any member of $F$ will be added correctly, thus eliminating an infinite family of inputs that need to be tested.

To perform the verification as described here, it is critical to construct $V$ correctly. If it is too small, then execution of the program on some input might require evaluation of $M(i)$ on some $i \notin V$, and so the behavior of $M(i)$ might deviate from $P(i)$. If it is too large, then the semantics of $P$ might not be well-defined on some elements in $V$, or the spurious step input sequences may not be reachable from any valid problem input (e.g., an array for quicksort or a DAG for topological sort).

To construct this set, by using the reference implementation of each subprogram, we construct a mapping between two sets of environment observations: the first set consists of all observations that can occur at the beginning of a particular subprogram's invocation, and the second set contains the observations at the end of that subprogram. We can obtain this mapping by first considering the possible observations that can arise at the beginning of the entry function (ADD, BUBBLE-SORT, TOPOSORT, and QUICKSORT) for some valid program input, and iteratively applying the observation-to-observation mapping implied by the reference implementation's step output at that point in the execution. If the step output specifies a primitive function call, we need to reason about how it can affect the environment so as to change the observation in the next step input. For non-primitive subprograms, we can update the observation-to-observation mapping currently associated with the subprogram and then apply that mapping to the current set. By iterating with this procedure, and then running $P$ on the input observation set that we obtain for the entry point function, we can obtain $V$ precisely. To make an analogy to MDPs, this procedure is analogous to how value iteration obtains the correct value for each state starting from any initialization.

An alternative method is to run $P$ on many different program inputs and then observe step input sequences which occur, to create $V$. However, to be sure that the generated $V$ is complete (covers all the cases needed), we need to check all pairs of observations seen in adjacent step inputs (in particular, those before and after a primitive function call), in a similar way as if we were constructing $V$ from scratch. Given a precise definition of $P$, it may be possible to automate the generation of $V$ from $P$ in future work.

Note that $V$ should also contain the necessary reductions, which corresponds to making the recursive calls at the correct time, as indicated by $P$.

After finding $V$, we construct a set of problem inputs which, when executed on $P$, create exactly the step input sequences which make up $V$. We call this set of inputs the *verification set*, $S_V$.

Given a verification set, we can then run the model on the verification set to check if the produced traces and results are correct. If yes, then this indicates that the learned neural program achieves provably perfect generalization.

We note that for tasks with very large input domains, such as ones involving MNIST digits or speech samples, the state space of base cases and reduction rules could be prohibitively large, possibly infinite. Consequently, it is infeasible to construct a verification set that covers all cases, and the verification procedure we have described is inadequate. We leave this as future work to devise a verification procedure more appropriate to this setting.

## 4 EXPERIMENTS

As there is no public implementation of NPI, we implemented a version of it in Keras that is as faithful to the paper as possible. Our experiments use a small number of training examples.

**Training Setup.** The training set for addition contains 200 traces. The maximum problem length in this training set is 3 (e.g., the trace corresponding to the problem "109 + 101").

The training set for bubble sort contains 100 traces, with maximum problem length of 2 (e.g., the trace corresponding to the array [3,2]).

The training set for topological sort contains 6 traces, with one synthesized from a graph of size 5 and the rest synthesized from graphs of size 7.

The training set for quicksort contains 4 traces, synthesized from arrays of length 5.

The same set of problems was used to generate the training traces for all formulations of the task, for non-recursive and recursive versions.

Table 1: Accuracy on Randomly Generated Problems for Bubble Sort

| Length of Array | Non-Recursive | Partially Recursive | Full Recursive |
|---|---|---|---|
| 2 | 100% | 100% | 100% |
| 3 | 6.7% | 23% | 100% |
| 4 | 10% | 10% | 100% |
| 8 | 0% | 0% | 100% |
| 20 | 0% | 0% | 100% |
| 90 | 0% | 0% | 100% |

We train using the Adam optimizer and use a 2-layer LSTM and task-specific state encoders for the external environments, as described in Reed & de Freitas (2016).

## 4.1 RESULTS ON GENERALIZATION OF RECURSIVE NEURAL PROGRAMS

We now report on generalization for the varying tasks.

**Grade-School Addition.** Both the non-recursive and recursive learned programs generalize on all input lengths we tried, up to 5000 digits. This agrees with the generalization of non-recursive addition in Reed & de Freitas (2016), where they reported generalization up to 3000 digits. However, note that there is no provable guarantee that the non-recursive learned program will generalize to all inputs, whereas we show later that the recursive learned program has a provable guarantee of perfect generalization.

In order to demonstrate that recursion can help learn and generalize better, for addition, we trained only on traces for 5 arbitrarily chosen 1-digit addition sum examples. The recursive version can generalize perfectly to long problems constructed from these components (such as the sum "822+233", where "8+2" and "2+3" are in the training set), but the non-recursive version fails to sum these long problems properly.

**Bubble Sort.** Table 1 presents results on randomly generated arrays of varying length for the learned non-recursive, partially recursive, and full recursive programs. For each length, we test each program on 30 randomly generated problems. Observe that partially recursive does slightly better than non-recursive for the setting in which the length of the array is 3, and that the fully recursive version is able to sort every array given to it. The non-recursive and partially recursive versions are unable to sort long arrays, beyond length 8.

**Topological Sort.** Both the non-recursive and recursive learned programs generalize on all graphs we tried, up to 120 vertices. As before, the non-recursive learned program lacks a provable guarantee of generalization, whereas we show later that the recursive learned program has one.

In order to demonstrate that recursion can help learn and generalize better, we trained a non-recursive and recursive model on just a single execution trace generated from a graph containing 5 nodes[3] for the topological sort task. For these models, Table 2 presents results on randomly generated DAGs of varying graph sizes (varying in the number of vertices). For each graph size, we test the learned programs on 30 randomly generated DAGs. The recursive version of topological sort solves all graph instances we tried, from graphs of size 5 through 70. On the other hand, the non-recursive version has low accuracy, beginning from size 5, and fails completely for graphs of size 8 and beyond.

**Quicksort.** Table 3 presents results on randomly generated arrays of varying length for the learned non-recursive and recursive programs. For each length, we test each program on 30 randomly generated problems. Observe that the non-recursive program's correctness degrades for length 11 and beyond, while the recursive program can sort any given array.

---

[3]The corresponding edge list is [(1, 2), (1, 5), (2, 4), (2, 5), (3, 5)].

Table 2: Accuracy on Randomly Generated Problems for Topological Sort

| Number of Vertices | Non-Recursive | Recursive |
|---|---|---|
| 5 | 6.7% | 100% |
| 6 | 6.7% | 100% |
| 7 | 3.3% | 100% |
| 8 | 0% | 100% |
| 70 | 0% | 100% |

Table 3: Accuracy on Randomly Generated Problems for Quicksort

| Length of Array | Non-Recursive | Recursive |
|---|---|---|
| 3 | 100% | 100% |
| 5 | 100% | 100% |
| 7 | 100% | 100% |
| 11 | 73.3% | 100% |
| 15 | 60% | 100% |
| 20 | 30% | 100% |
| 22 | 20% | 100% |
| 25 | 3.33% | 100% |
| 30 | 3.33% | 100% |
| 70 | 0% | 100% |

As mentioned in Section 2.1, we hypothesize the non-recursive programs do not generalize well because they have learned spurious dependencies specific to the training set, such as length of the input problems. On the other hand, the recursive programs have learned the true program semantics.

## 4.2 VERIFICATION OF PROVABLY PERFECT GENERALIZATION

We describe how models trained with recursive traces can be proven to generalize, by using the verification procedure described in Section 3.3. As described in the verification procedure, it is possible to prove our learned recursive program generalizes perfectly by testing on an appropriate set of problem inputs, i.e., the verification set. Recall that this verification procedure cannot be performed for the non-recursive versions, since the propagation of the hidden state in the core LSTM module makes reasoning difficult and so we would need to check an unbounded number of examples.

We describe the base cases, reduction rules, and the verification set for each task in Appendix A.6. For each task, given the verification set, we check the traces and results of the learned, to-be-verified neural program (described in Section 4.1; and for bubble sort, Appendix A.6) on the verification set, and ensure they *match* the traces produced by the true program $P$. Our results show that for all learned, to-be-verified neural programs, they all produced the same traces as those produced by $P$ on the verification set. Thus, we demonstrate that recursion enables provably perfect generalization for different tasks, including addition, topological sort, quicksort, and a variant of bubble sort.

Note that the training set can often be considerably smaller than the verification set, and despite this, the learned model can still pass the entire verification set. Our result shows that the training procedure and the NPI architecture is capable of generalizing from the step input-output pairs seen in the training data to the unseen ones present in the verification set.

## 5 CONCLUSION

We emphasize that the notion of a neural recursive program has not been presented in the literature before: this is our main contribution. Recursion enables provably perfect generalization. To the best of our knowledge, this is the first time verification has been applied to a neural program, provid-

ing provable guarantees about its behavior. We instantiated recursion for the Neural Programmer-Interpreter by changing the training traces. In future work, we seek to enable more tasks with recursive structure. We also hope to decrease supervision, for example by training with only partial or non-recursive traces, and to develop novel Neural Programming Architectures integrated directly with a notion of recursion.

ACKNOWLEDGMENTS

This material is in part based upon work supported by the National Science Foundation under Grant No. TWC-1409915, DARPA under Grant No. FA8750-15-2-0104, and Berkeley Deep Drive. Any opinions, findings, and conclusions or recommendations expressed in this material are those of the author(s) and do not necessarily reflect the views of National Science Foundation and DARPA.

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

# A  APPENDIX

## A.1  PROGRAM SET FOR NON-RECURSIVE TOPOLOGICAL SORT

| Program | Descriptions | Calls | Arguments |
|---------|-------------|-------|-----------|
| TOPOSORT | Perform topological sort on graph | TRAVERSE, NEXT_START, WRITE, MOVE | NONE |
| TRAVERSE | Traverse graph until stack is empty | CHECK_CHILD, EXPLORE | NONE |
| CHECK_CHILD | Check if a white child exists; if so, set $childList[v_{active}]$ to point to it | MOVE | NONE |
| EXPLORE | Repeatedly traverse subgraphs until stack is empty | STACK, CHECK_CHILD, WRITE, MOVE | NONE |
| STACK | Interact with stack, either pushing or popping | WRITE, MOVE | PUSH, POP |
| NEXT_START | Move $p_{start}$ until reaching a white vertex. If a white vertex is found, set $p_{start}$ to point to it; this signifies the start of a traversal of a new connected component. If no white vertex is found, the entire execution is terminated | MOVE | NONE |
| WRITE | Write a value either to environment (e.g., to color a vertex) or variable (e.g., to change the value of $v_{active}$) | NONE | Described below |
| MOVE | Move a pointer (e.g., $p_{start}$ or $childList[v_{active}]$) up or down | NONE | Described below |

**Argument Sets for WRITE and MOVE.**

**WRITE.**  The WRITE operation has the following arguments:

*ARG_1* (Main Action): COLOR_CURR, COLOR_NEXT, ACTIVE_START, ACTIVE_NEIGHB, ACTIVE_STACK, SAVE, STACK_PUSH, STACK_POP, RESULT

COLOR_CURR colors $v_{active}$, COLOR_NEXT colors Vertex $DAG[v_{active}][childList[v_{active}]]$, ACTIVE_START writes $p_{start}$ to $v_{active}$, ACTIVE_NEIGHB writes $DAG[v_{active}][childList[v_{active}]]$ to $v_{active}$, ACTIVE_STACK writes $Q_{stack}(p_{stack})$ to $v_{active}$, SAVE writes $v_{active}$ to $v_{save}$, $STACK\_PUSH$ pushes $v_{active}$ to the top of the stack, $STACK\_POP$ writes a null value to the top of the stack, and $RESULT$ writes $v_{active}$ to $Q_{result}(p_{result})$.

*ARG_2* (Auxiliary Variable): COLOR_GREY, COLOR_BLACK

COLOR_GREY and COLOR_BLACK color the given vertex grey and black, respectively.

**MOVE.**  The MOVE operation has the following arguments:

*ARG_1* (Pointer): $p_{result}, p_{stack}, p_{start}, childList[v_{active}], childList[v_{save}]$

Note that the argument is the identity of the pointer, not what the pointer points to; in other words, *ARG_1* can only take one of 5 values.

*ARG_2* (Increment or Decrement): UP, DOWN

## A.2  TRACE-GENERATING FUNCTIONS FOR TOPOLOGICAL SORT

### A.2.1  NON-RECURSIVE TRACE-GENERATING FUNCTIONS

```
1    // Top level topological sort call
2    TOPOSORT() {
3      while (Q_color(p_start) is a valid color): // color invalid when all vertices explored
4        WRITE(ACTIVE_START)
5        WRITE(COLOR_CURR, COLOR_GREY)
6        TRAVERSE()
7        MOVE(p_start, UP)
8        NEXT_START()
9    }
10
11   TRAVERSE() {
12     CHECK_CHILD()
13     EXPLORE()
14   }
15
16   CHECK_CHILD() {
17     while (Q_color(DAG[v_active][childList[v_active]]) is not white and is not invalid): // color invalid when all children explored
18       MOVE(childList[v_active], UP)
19   }
20
21   EXPLORE() {
22     do
23       if (Q_color(DAG[v_active][childList[v_active]]) is white):
24         WRITE(COLOR_NEXT, COLOR_GREY)
25         STACK(PUSH)
26         WRITE(SAVE)
27         WRITE(ACTIVE_NEIGHB)
28         MOVE(childList[v_save], UP)
29       else:
30         WRITE(COLOR_CURR, COLOR_BLACK)
31         WRITE(RESULT)
32         MOVE(p_result, UP)
33         if(p_stack == 1):
34           break
35         else:
36           STACK(POP)
37         CHECK_CHILD()
38     while (true)
39   }
40
41   STACK(op) {
42     if (op == PUSH):
43       WRITE(STACK_PUSH)
44       MOVE(p_stack, UP)
45
46     if (op == POP):
47       WRITE(ACTIVE_STACK)
48       WRITE(STACK_POP)
49       MOVE(p_stack, DOWN)
50   }
51
52   NEXT_START() {
53     while(Q_color(p_start) is not white and is not invalid): // color invalid when all vertices explored
54       MOVE(p_start, UP)
55   }
```

### A.2.2 RECURSIVE TRACE-GENERATING FUNCTIONS

Altered Recursive Functions

```
1    // Top level topological sort call
2    TOPOSORT() {
3      if (Q_color(p_start) is a valid color): // color invalid when all vertices explored
4        WRITE(ACTIVE_START)
5        WRITE(COLOR_CURR, COLOR_GREY)
6        TRAVERSE()
7        MOVE(p_start, UP)
8        NEXT_START()
9        TOPOSORT() // Recursive Call
10   }
11
12   CHECK_CHILD() {
13     if (Q_color(DAG[v_active][childList[v_active]]) is not white and is not invalid): // color invalid when all children explored
14       MOVE(childList[v_active], UP)
15       CHECK_CHILD() // Recursive Call
16   }
17
18   EXPLORE() {
19     if (Q_color(DAG[v_active][childList[v_active]]) is white):
20       WRITE(COLOR_NEXT, COLOR_GREY)
21       STACK(PUSH)
22       WRITE(SAVE)
23       WRITE(ACTIVE_NEIGHB)
24       MOVE(childList[v_save], UP)
25     else:
26       WRITE(COLOR_CURR, COLOR_BLACK)
27       WRITE(RESULT)
28       MOVE(p_result, UP)
29       if(p_stack == 1):
30         return
31       else:
32         STACK(POP)
33     CHECK_CHILD()
34     EXPLORE() // Recursive Call
35   }
36
37   NEXT_START() {
38     if (Q_color(p_start) is  not white and is not invalid): // color invalid when all vertices explored
39       MOVE(p_start, UP)
40       NEXT_START() // Recursive Call
41   }
```

## A.3 NON-RECURSIVE QUICKSORT

---

**Algorithm 4** Iterative Quicksort

---

1: Initialize an array $A$ to sort and two empty stacks $S_{lo}$ and $S_{hi}$.
2: Initialize $lo$ and $hi$ to be 1 and $n$, where $n$ is the length of $A$.
3:
4: **function** PARTITION($A, lo, hi$)
5: $pivot = lo$
6: **for** $j \in [lo, hi-1]$ : **do**
7: **if** $A[j] \leq A[hi]$ **then**
8: swap $A[pivot]$ with $A[j]$
9: $pivot = pivot + 1$
10: swap $A[pivot]$ with $A[hi]$
11: return $pivot$

12:
13: **function** QUICKSORT($A, lo, hi$)
14: **while** $S_{lo}$ and $S_{hi}$ are not empty: **do**
15: Pop states off $S_{lo}$ and $S_{hi}$, writing them to $lo$ and $hi$.
16: p = PARTITION($A, lo, hi$)
17: Push $p + 1$ and $hi$ to $S_{lo}$ and $S_{hi}$.
18: Push $lo$ and $p - 1$ to $S_{lo}$ and $S_{hi}$.

---

A.4 PROGRAM SET FOR QUICKSORT

| Program | Descriptions | Calls | Arguments |
|---|---|---|---|
| QUICKSORT | Run the quicksort routine in place for the array $A$, for indices from $lo$ to $hi$ | **Non-Recursive**: PAR-TITION, STACK, WRITE **Recursive**: same as non-recursive version, along with QUICK-SORT | Implicitly: array $A$ to sort, $lo$, $hi$ |
| PARTITION | Runs the partition function. At end, pointer $p_{pivot}$ is moved to the pivot | COMPSWAP_LOOP, MOVE_PIVOT_LO, MOVE_J_LO, SWAP | NONE |
| COMPSWAP_LOOP | Runs the FOR loop inside the partition function | COMPSWAP, MOVE | NONE |
| COMPSWAP | Compares $A[pivot] \leq A[j]$; if so, perform a swap and increment $p_{pivot}$ | SWAP, MOVE | NONE |
| SET_PIVOT_LO | Sets $p_{pivot}$ to $lo$ index | NONE | NONE |
| SET_J_LO | Sets $p_j$ to $lo$ index | NONE | NONE |
| SET_J_NULL | Sets $p_j$ to $-\infty$ | NONE | NONE |
| STACK | Pushes $lo/hi$ states onto stacks $S_{lo}$ and $S_{hi}$ according to argument (described below) | WRITE, MOVE | Described below |
| MOVE | Moves pointer one unit up or down | NONE | Described below |
| SWAP | Swaps elements at given array indices | NONE | Described below |
| WRITE | Write a value either to stack (e.g., $Q_{stackLo}$ or $Q_{stackHi}$) or to pointer (e.g., to change the value of $p_{hi}$) | NONE | Described below |

**Argument Sets for STACK, MOVE, SWAP, WRITE.**

**STACK.** The STACK operation has the following arguments:

*ARG_1* (Operation): STACK_PUSH_CALL1, STACK_PUSH_CALL2, STACK_POP

STACK_PUSH_CALL1 pushes $lo$ and $pivot - 1$ to $Q_{stackLo}$ and $Q_{stackHi}$. STACK_PUSH_CALL2 pushes $pivot + 1$ and $hi$ to $Q_{stackLo}$ and $Q_{stackHi}$. STACK_POP pushes $-\infty$ values to $Q_{stackLo}$ and $Q_{stackHi}$.

**MOVE.** The MOVE operation has the following arguments:

*ARG_1* (Pointer): $p_{stackLo}, p_{stackHi}, p_j, p_{pivot}$

Note that the argument is the identity of the pointer, not what the pointer points to; in other words, *ARG_1* can only take one of 4 values.

*ARG_2* (Increment or Decrement): UP, DOWN

**SWAP.** The SWAP operation has the following arguments:

*ARG_1* (Swap Object 1): $p_{pivot}$

*ARG_2* (Swap Object 2): $p_{hi}, p_j$

**WRITE.** The WRITE operation has the following arguments:

*ARG_1* (Object to Write): ENV_STACK_LO, ENV_STACK_HI, $p_{hi}, p_{lo}$

ENV_STACK_LO and ENV_STACK_HI represent $Q_{stackLo}(p_{stackLo})$ and $Q_{stackHi}(p_{stackHi})$, respectively.

*ARG_2* (Object to Copy): ENV_STACK_LO_PEEK, ENV_STACK_HI_PEEK, $p_{hi}, p_{lo}, p_{pivot} - 1$, $p_{pivot} + 1$, RESET

ENV_STACK_LO_PEEK and ENV_STACK_HI_PEEK represent $Q_{stackLo}(p_{stackLo} - 1)$ and $Q_{stackHi}(p_{stackHi} - 1)$, respectively. RESET represents a $-\infty$ value.

Note that the argument is the identity of the pointer, not what the pointer points to; in other words, *ARG_1* can only take one of 4 values, and *ARG_2* can only take one of 7 values.

## A.5  TRACE-GENERATING FUNCTIONS FOR QUICKSORT

### A.5.1  NON-RECURSIVE TRACE-GENERATING FUNCTIONS

```
1   Initialize p_lo to 1 and p_hi to n (length of array)
2   Initialize p_j to -∞
3
4   QUICKSORT() {
5     while (p_stackLo ≠ 1):
6       if (Q_stackLo(p_stackLo - 1) < Q_stackHi(p_stackHi - 1)):
7         STACK(STACK_POP)
8       else:
9         WRITE(p_hi, ENV_STACK_HI_PEEK)
10        WRITE(p_lo, ENV_STACK_LO_PEEK)
11        STACK(STACK_POP)
12        PARTITION()
13        STACK(STACK_PUSH_CALL2)
14        STACK(STACK_PUSH_CALL1)
15  }
16
17  PARTITION() {
18    SET_PIVOT_LO()
19    SET_J_LO()
20    COMPSWAP_LOOP()
21    SWAP(p_pivot, p_hi)
22    SET_J_NULL()
23  }
24
25  COMPSWAP_LOOP() {
26    while (p_j ≠ p_hi):
27      COMPSWAP()
28      MOVE(p_j, UP)
29  }
30
31  COMPSWAP() {
32    if (A[p_j] ≤ A[p_hi]):
33      SWAP(p_pivot, p_j)
34      MOVE(p_pivot, UP)
35  }
36
37  STACK(op) {
38    if (op == STACK_PUSH_CALL1):
39      WRITE(ENV_STACK_LO, p_lo)
40      WRITE(ENV_STACK_HI, p_pivot - 1)
41      MOVE(p_stackLo, UP)
42      MOVE(p_stackHi, UP)
43
44    if (op == STACK_PUSH_CALL2):
45      WRITE(ENV_STACK_LO, p_pivot + 1)
46      WRITE(ENV_STACK_HI, p_hi)
47      MOVE(p_stackLo, UP)
48      MOVE(p_stackHi, UP)
49
50    if (op == STACK_POP):
51      WRITE(ENV_STACK_LO, RESET)
52      WRITE(ENV_STACK_HI, RESET)
53      MOVE(p_stackLo, DOWN)
54      MOVE(p_stackHi, DOWN)
55  }
```

### A.5.2 RECURSIVE TRACE-GENERATING FUNCTIONS

Altered Recursive Functions

```
1    Initialize p_lo to 1 and p_hi to n (length of array)
2    Initialize p_j to -∞
3
4    QUICKSORT() {
5      if (Q_stackLo(p_stackLo - 1) < Q_stackHi(p_stackHi - 1)):
6        PARTITION()
7        STACK(STACK_PUSH_CALL2)
8        STACK(STACK_PUSH_CALL1)
9        WRITE(p_hi, ENV_STACK_HI_PEEK)
10       WRITE(p_lo, ENV_STACK_LO_PEEK)
11       QUICKSORT() // Recursive Call
12       STACK(STACK_POP)
13       WRITE(p_hi, ENV_STACK_HI_PEEK)
14       WRITE(p_lo, ENV_STACK_LO_PEEK)
15       QUICKSORT() // Recursive Call
16       STACK(STACK_POP)
17    }
18
19   COMPSWAP_LOOP() {
20      if (p_j ≠ p_hi):
21        COMPSWAP()
22        MOVE(p_j, UP)
23        COMPSWAP_LOOP() // Recursive Call
24   }
```

### A.6 BASE CASES, REDUCTION RULES, AND VERIFICATION SETS

In this section, we describe the space of base cases and reduction rules that must be covered for each of the four sample tasks, in order to create the verification set.

For addition, we analytically determine the verification set. For tasks other than addition, it is difficult to analytically determine the verification set, so instead, we randomly generate input candidates until they completely cover the base cases and reduction rules.

**Base Cases and Reduction Rules for Addition.**   For the recursive formulation of addition, we analytically construct the set of input problems that cover all base cases and reduction rules. We outline how to construct this set.

It is sufficient to construct problems where every transition between two adjacent columns is covered. The ADD reduction rule ensures that each call to ADD only covers two adjacent columns, and so the LSTM only ever runs for a fixed number of steps necessary to process these two columns.

We construct input problems by splitting into two cases: one case in which the left column contains a null value and another in which the left column does not contain any null values. We then construct problem configurations that span all possible valid environment states (for instance, in order to force the carry bit in a column to be 1, one can add the sum "1+9" in the column to the right).

The operations we need to be concerned most about are CARRY and LSHIFT, which induce *partial environment states* spanning two columns. It is straightforward to deal with all other operations, which do not induce partial environment states.

Under the assumption that there are no leading 0's (except in the case of single digits) and the two numbers to be added have the same number of digits, the verification set for addition contains 20,181 input problems. The assumption of leading 0's can be easily removed, at the cost of slightly increasing the size of the verification set. We made the assumption of equivalent lengths in order to parametrize the input format with respect to length, but this assumption can be removed as well.

**Base Cases and Reduction Rules for Bubble Sort.**   The original version of the bubblesort implementation exposes the values within the array. While this matches the description from Reed & de Freitas (2016), we found that this causes an unnecessary blowup in the size of $V$ and makes it much more difficult to construct the verification set. For purposes of verification, we replace the domain-specific encoder with the following:

$$f_{enc}(Q, i_1, i_2, i_3, a_t) = MLP([Q(1, i_1) \leq Q(1, i_2), 1 \leq i_1 \leq length, 1 \leq i_2 \leq length,$$
$$i_3 == length, a_t(1), a_t(2), a_t(3)]),$$

Table 4: Accuracy on Randomly Generated Problems for Variant of Bubble Sort

| Length of Array | Non-Recursive | Recursive |
|---|---|---|
| 2 | 100% | 100% |
| 3 | 100% | 100% |
| 4 | 100% | 100% |
| 5 | 100% | 100% |
| 6 | 90% | 100% |
| 7 | 86.7% | 100% |
| 8 | 6.7% | 100% |
| 9 | 0% | 100% |
| 10 | 0% | 100% |
| 12 | 0% | 100% |
| 15 | 0% | 100% |
| 70 | 0% | 100% |

which directly exposes which of the two values pointed to is larger. This modification also enables us to sort arrays containing arbitrary comparable elements.

By reasoning about the possible set of environment observations created by all valid inputs, we construct $V$ using the procedure described in Section 3.3. Using this modification, we constructed a verification set consisting of one array of size 10.

We also report on generalization results for the non-recursive and recursive versions of this variant of bubble sort. Table 4 demonstrates that the accuracy of the non-recursive program degrades sharply when moving from arrays of length 7 to arrays of length 8. This is due to the properties of the training set – we trained on 2 traces synthesized from arrays of length 7 and 1 trace synthesized from an array of length 6. Table 4 also demonstrates that the (verified) recursive program generalizes perfectly.

**Base Cases and Reduction Rules for Topological Sort.** For each function we use to implement the recursive version of topological sort, we need to consider the set of possible environment observation sequences we can create from all valid inputs and test that the learned program produces the correct behavior on each of these inputs. We have three observations: the color of the start node, the color of the active node's next child to be considered, and whether the stack is empty. Naïvely, we might expect to synthesize and test an input for any sequence created by combining the four possible colors in two variables and another boolean variable for whether the stack is empty (so 32 possible observations at any point), but for various reasons, most of these combinations are impossible to occur at any given point in the execution trace.

Through careful reasoning about the possible set of environment observations created by all valid inputs, and how each of the operations in the execution trace affects the environment, we can construct $V$ using the procedure described in Section 3.3. We then construct a verification set of size 73 by ensuring that randomly generated graphs cover the analytically derived $V$. The model described in the training setup of Section 4 (trained on 6 traces) was verified to be correct via the matching procedure described in Section 4.2.

**Base Cases and Reduction Rules for Quicksort.** As with the others, we apply the procedure described in Section 3.3 to construct $V$ and then empirically create a verification set which covers $V$. The verification set can be very small, as we found a 10-element array ([8,2,1,2,0,8,5,8,3,7]) is sufficient to cover all of $V$. We note that an earlier version of quicksort we tried lacked primitive operations to directly move a pointer to another, and therefore needed more functions and observations. As this complexity interfered with determining the base cases and reductions, we changed the algorithm to its current form. Even though the earlier version also generalized just as well in practice, relatively small differences in the formulation of the traces and the environment observations can drastically change the difficulty of verification.

