# Peer review of "Making Neural Programming Architectures Generalize via Recursion"

_ICLR 2017 — accepted_

[Official Review · AnonReviewer1 · rating 8 · confidence 4 · 16 Dec 2016]
**A demonstration that NPI can learn to solve Tower of Hanoi!**
originality 4

This paper argues that being able to handle recursion is very important for neural programming architectures — that handling recursion allows for strong generalization to out of domain test cases and learning from smaller amounts of training data.  Most of the paper is a riff on the Reed & de Freitas paper on Neural Programmer Interpreters from ICLR 2016 which learns from program traces — this paper trains NPI models on traces that have recursive calls.  The authors show how to verify correctness by evaluating the learned program on only a small set of base cases and reduction rules and impressively, show that the NPI architecture is able to perfectly infer Bubblesort and the Tower of Hanoi problems.  

What I like is that the idea is super simple and as the authors even mention, the only change is to the execution traces that the training pipeline gets to see.  I’m actually not sure what the right take-away is — does this mean that we have effectively solved the neural programming problem when the execution traces are available? (and was the problem too easy to begin with?).    For example, a larger input domain (as one of the reviewers also mentions) is MNIST digits and we can imagine a problem where the NPI must infer how to sort MNIST digits from highest to lowest.  In this setting, having execution traces would effectively decouple the problem of recognizing the digits from that of inferring the program logic — and so the problem would be no harder than learning to recognize MNIST digits and learning to bubble sort from symbols.  What is a problem where we have access to execution traces but cannot infer it using the proposed method?

[Official Review · AnonReviewer2 · rating 8 · confidence 3 · 16 Dec 2016]
**Nifty extension to make NPI more practical**

This is a very interesting and fairly easy to read paper. 
The authors present a small, yet nifty approach to make Neural Programming Interpreters significantly more powerful. By allowing recursion, NPI generalizes better from fewer execution traces.
It's an interesting example of how a small but non-trivial extension can make a machine learning method significantly more practical.

I also appreciate that the same notation was used in this paper and the original Deepmind paper. As a non-expert on this topic, it was easy to read the original paper in tandem. 

My one point of critique is that the generalization proves are a bit vague. For the numerical examples in the paper, you can iterate over all possible execution paths until the next recursive call. However, how would this approach generalize a continuous input space (e.g. the 3D car example in the original paper). It seems that a prove of generalization will still be intractable in the continuous case? 

Are you planning on releasing the source code?

[Official Review · AnonReviewer3 · rating 9 · confidence 5 · 16 Dec 2016]
**Greatly improved training and analysis of NPI**
soundness 2 · originality 2 · clarity 3 · meaningful comparison 2

This paper improves significantly upon the original NPI work, showing that the model generalizes far better when trained on traces in recursive form. The authors show better sample complexity and generalization results for addition and bubblesort programs, and add two new and more interesting tasks - topological sort and quicksort (added based on reviewer discussion). Furthermore, they actually *prove* that the algorithms learned by the model generalize perfectly, which to my knowledge is the first time this has been done in neural program induction.

[Public Comment · (anonymous) · 02 May 2017]
**Source code**

Has the source code for this paper been released by the authors?

[Final Decision · Program Chairs · 06 Feb 2017]
**ICLR committee final decision**

The reviewers were very favourable, and the paper is on a highly-relevant topic and explores a useful practical trick.